# Tribological Properties of Cr$_2$O$_3$, Cr$_2$O$_3$–SiO$_2$-TiO$_2$ and Cr$_2$O$_3$–SiO$_2$-TiO$_2$-Graphite Coatings Deposited by Atmospheric Plasma Spraying

Lukas Bastakys [1,*], Liutauras Marcinauskas [1,2], Mindaugas Milieška [2], Mitjan Kalin [3] and Romualdas Kėželis [2]

[1] Department of Physics, Kaunas University of Technology, Studentų Str. 50, LT-51368 Kaunas, Lithuania
[2] Plasma Processing Laboratory, Lithuanian Energy Institute, Breslaujos Str. 3, LT-44403 Kaunas, Lithuania
[3] Laboratory for Tribology and Interface Nanotechnology, Faculty of Mechanical Engineering, University of Ljubljana, Bogišićeva 8, 1000 Ljubljana, Slovenia
* Correspondence: lukas.bastakys@ktu.lt or lukas.bastakys@ktu.edu; Tel.: +370-6763-3489

**Abstract:** In this study, Cr$_2$O$_3$, Cr$_2$O$_3$-SiO$_2$-TiO$_2$ and Cr$_2$O$_3$-SiO$_2$-TiO$_2$-graphite coatings were formed by atmospheric plasma spraying. The influence of SiO$_2$-TiO$_2$ and SiO$_2$-TiO$_2$-graphite reinforcements on the surface morphology, elemental composition, structure and tribological properties of chromia coatings was determined. The friction coefficients and specific wear rates were investigated by a ball-on-flat configuration using 1 N and 3 N loads under dry-lubrication conditions. The addition of SiO$_2$-TiO$_2$-graphite resulted in the lowest surface roughness and the most homogenous surface of the coatings. The X-ray diffraction (XRD) measurements demonstrated that all as-sprayed coatings consisted of an eskolaite chromium oxide phase. The results showed that the Cr$_2$O$_3$-SiO$_2$-TiO$_2$ coating demonstrated the lowest friction coefficient values. The SiO$_2$-TiO$_2$ and SiO$_2$-TiO$_2$-graphite additives reduced the specific wear rates of Cr$_2$O$_3$ coatings by 30% and 45%, respectively. Additionally, the wear resistance was improved almost 45 times in comparison to the steel substrate.

**Keywords:** plasma spraying; chromia coatings; graphite; tribological properties; friction coefficient





## 1. Introduction

Cr$_2$O$_3$ coatings are widely used in various fields in order to reduce the wear of metallic parts and metallic surfaces from corrosion [1–6].

Plasma spraying is one of the commonly used techniques to deposit ceramic coatings, such as aluminum oxide, chromium oxide, titanium oxide, zirconium oxide, etc., on metallic surfaces [2,4,7–10]. The mechanical, tribological, thermal or corrosion properties of sprayed Cr$_2$O$_3$ coatings are easily controlled by changing the spraying conditions such as spraying distance, arc current, gas type, gas flow ratio, etc. [2,8,11,12]. Kiilakoski et al. [8] indicated that the surface roughness, hardness, porosity and wear resistance of chromia coatings could be controlled by changing the spray distance, flow rate of air or suspension feed rate.

Another way to improve the hardness and toughness, reduce the friction coefficient, or enhance the wear and corrosion resistance of chromia coatings is to use additive materials. It was demonstrated that TiO$_2$, Al$_2$O$_3$, YSZ-SiC or CeO$_2$ is used to improve the properties of Cr$_2$O$_3$ coatings [2,13–18]. Li et al. [14] demonstrated that the addition of TiO$_2$ up to 32 wt.% will improve the tribological properties under dry friction and oil friction conditions. However, it was found that the friction coefficient of Cr$_2$O$_3$-TiO$_2$ coatings in most cases was enhanced from 0.60 up to 0.92 under dry sliding. Our previous investigations [15] indicated that the friction coefficient was reduced, and the specific wear rate decreased from $2.69 \times 10^{-6}$ mm$^3$/(Nm) to $1.54 \times 10^{-6}$ mm$^3$/(Nm) after doping of chromium oxide coating with SiO$_2$-TiO$_2$. Mao et al. [16] observed that the wear rate of the chromium oxide coating was $14.6 \times 10^{-7}$ mm$^3$/(Nm) and, depending on the alumina fraction, varied from 8.2 to $12.9 \times 10^{-7}$ mm3/(Nm) for Cr$_2$O$_3$-Al$_2$O$_3$ coatings. However, the friction coefficient

values under dry-sliding conditions for $Cr_2O_3$ and $Cr_2O_3$-$Al_2O_3$ coatings remained similar. Bolelli et al. [17] demonstrated that the friction coefficients and wear rates were enhanced with the addition of $TiO_2$, $Al_2O_3$ or $ZrO_2$ into $Cr_2O_3$ coatings. Singh et al. [19] showed that the friction coefficient of conventional and nanostructured $Cr_2O_3$-3%$TiO_2$ coatings varied between 0.55 and 0.80 depending on the applied loads and sliding distances. Ding et al. [20] showed that the properties of $Cr_2O_3$-20wt.% $TiO_2$ coatings could be improved with the addition of $CeO_2$. It was obtained that the surface roughness and the porosity were reduced, the bonding strength was enhanced and the friction coefficient was reduced from 0.68 to 0.62 with the addition of $CeO_2$ into $Cr_2O_3$-$TiO_2$ coatings. Hashemi et al. [21] obtained that $Cr_2O_3$-YSZ and $Cr_2O_3$-YSZ-SiC coatings demonstrated higher porosity, lower bonding strength and slightly higher friction coefficient values, but the wear rate of the coatings was reduced up to three times compared to the $Cr_2O_3$ coating.

The tribological properties of plasma-sprayed ceramic coatings could be improved with the addition of solid lubricant materials such as graphene nanoplatelets [22,23], carbon nanotubes [24], graphene oxide [25] or graphite [26,27]. It was demonstrated that graphene oxide improved the fracture toughness and reduced the microhardness of alumina-graphene oxide-sprayed coatings [25]. The addition of both carbon nanotubes (CNTs) and graphite nanoparticles (GNPs) increased the hardness, elastic modulus and fracture toughness of alumina coatings [23]. Venturi et al. [22] observed that the wear rate was reduced by ~20%, while the friction coefficient decreased from 0.60 to 0.51 with the incorporation of graphene nanoplatelets into $Cr_2O_3$ coatings. Bagde et al. [27] indicated that the addition of Ni-graphite into the $Cr_2O_3$-$TiO_2$ coating decreased the friction coefficient values, while the abrasive wear resistance was enhanced from 1.3 up to 2 times depending on the used loads. Goyal et al. [24] showed that the reinforcement of chromia coatings with carbon nanotubes drastically reduced corrosion rates. The performed studies demonstrated that the incorporation of $TiO_2$ improves the mechanical and tribological properties of the $Cr_2O_3$ coatings, but the properties of the coatings are greatly influenced by the concentration of $TiO_2$, type of used powder and spraying conditions [13–15,17,19]. Another effective way to improve the tribological properties of ceramic coatings is to introduce carbon-based materials (graphite, graphene, graphene oxide, etc.) into the ceramic matrix [22–27]. It should be noted that studies on the incorporation of carbon-based materials into chromia coatings are very scarce in the scientific literature. In addition, the effect of graphite on the tribological behavior of chromium oxide composite coatings formed by atmospheric plasma spraying using air-hydrogen plasma was not studied previously.

The main aim was to investigate the effect of the addition of $SiO_2$-$TiO_2$ and $SiO_2$-$TiO_2$-graphite on the surface morphology, phase structure, elemental composition and tribological properties under the dry sliding of chromium oxide composite coatings deposited by atmospheric plasma spraying.

## 2. Experimental Section

The plasma torch used for the formation of the coatings was developed at the Lithuanian Energy Institute [28]. The P265GH steel was used as a substrate to deposit the chromium ($Cr_2O_3$) and chromium composite ($Cr_2O_3$-$SiO_2$-$TiO_2$ and $Cr_2O_3$-$SiO_2$-$TiO_2$-graphite) coatings. The dimensions of the P265GH steel were: length, 40 mm; width, 10 mm; and thickness, 6 mm. The steel was located on a holder that was cooled by water. The deposition of the coatings was performed using atmospheric pressure air plasma spraying, and the parameters of the process were: arc current, 200 A; total gas flow rate, 3.7 g/s; hydrogen gas, which was used to enhance the degree of melting of the powder particles, 0.053 g/s. Such parameters resulted in ~41.0 kW of torch power and an average temperature of the plasma at the torch nozzle chamber (at the point of powder injection) of 3780 ± 50 K. The average plasma temperature at the torch nozzle exit was 3480 ± 30 K. Coatings were formed as the powders were injected into the plasma by a powder carrier gas (air) at a flow rate of 0.48 g/s. The distance between the plasma torch nozzle outlet and the coating surface was 70 mm, and the spraying duration was 40 s. The MOGUL PC 18

powder (MOGUL METALLIZING GmbH, Kottinbrunn, Austria) of $Cr_2O_3$ (purity, 99.7%; and mesh size, 45 + 22 μm) was used to deposit the chromium oxide coatings. Meanwhile, the $Cr_2O_3$ composite coatings were sprayed using $Cr_2O_3$-$SiO_2$-$TiO_2$ (92/5/3, MOGUL PC 17, mesh size: 45 + 22 m, MOGUL METALLIZING GmbH, Kottinbrunn, Austria) powders, in which $SiO_2$ and $TiO_2$ were mixed in 5% and 3% by weight, respectively [15]. Graphite powders of nonregular shape (MOLYDUVAL Fondra NS, particle size: <25 μm) were also used [29]. The graphite was mixed into the $Cr_2O_3$-$SiO_2$-$TiO_2$ at the weight ratio of 10% to produce $Cr_2O_3$-$SiO_2$-$TiO_2$-graphite feedstock powders. The powders were mechanically mixed for 24 hours. All feedstock powders were dried at ~350 K for at least 18 hours before the plasma spraying. For better adhesion between the coating and the substrate, Al as a bonding layer was deposited.

The top-view morphology of the deposited coatings was investigated by scanning electron microscopy (SEM) (Hitachi S-3400N (Hitachi, Tokyo, Japan)), and the elemental composition was measured using energy-dispersive X-ray spectroscopy (EDS) with a Bruker Quad 5040 spectrometer (AXS Microanalysis GmbH, Billerica, MA, USA). The measurement data were collected and measured from at least five different surface locations with a surface area magnification of x100. Linear surface roughness was measured using a Mitutoyo Surftest SJ-210 Series tester (version 2.00 with standard ISO 1997 Mitutoyo, Kawasaki, Japan). The roughness values given are the average of at least 15 separate measurements. The length of one track was 4 mm, the speed of the measurement was 0.50 mm/s, the detector measuring force was 0.75 mN and a diamond tip was used. The calculation of average roughness values ($R_a$, $R_q$) was performed by a roughness tester according to the ISO4287:1997 standard. Cut-off $\lambda_c$ was equal to 0.8 mm. The surface roughness of all coatings was measured at the same conditions.

The phase analysis was carried out by the X-ray diffraction technique (XRD) using the D8 Discover apparatus (Bruker D8 Discover, Billerica, MA, USA) with a CuK$\alpha$ ($\lambda$ = 0.154059 nm) cathode. The XRD patterns were recorded under standard Bragg–Brentano configuration in the range of 2θ from 10° to 80°. The diffraction patterns of the coatings were analyzed by DIFFRAC.EVA software. The tribological properties were determined using a CETR-UMT-2 tribometer (CETR, Campbell, CA, USA) and ball-on-flat configuration with 1 N and 3 N loads under dry-sliding conditions. The duration of the tribological tests was 120 min, the sliding speed was 0.1 m/s, and the total covered distance was 720 m. A 10 mm diameter $Al_2O_3$ ball (grade class 10 and 99.5%) was used as a counterpart, and the stroke length was 5 mm. The specific wear rate was calculated from the linear surface profiles that were obtained with Ambios XP-200 Profiler (Ambios Technology Inc., Santa Cruz, CA, USA). Tribological tests of $Cr_2O_3$ and chromia composite coatings were performed on two samples of each series, and average values (of at least four measurements) were determined.

## 3. Results and Discussion

The as-sprayed chromia and chromia composite coatings' surface morphology is shown in Figure 1. It was obtained that the deposited coatings consisted of melted particle regions with partially melted particles. It should be noted that the amount of fully melted particle regions in the $Cr_2O_3$-$SiO_2$-$TiO_2$ coating was slightly higher compared to the $Cr_2O_3$ coating (Figure 1b,d). Mostly, the surface of the $Cr_2O_3$-$SiO_2$-$TiO_2$-graphite coating was even and homogenous (Figure 1e,f). All deposited coatings had a grain structure with a low amount of microcracks and pores (Figure 1). The existence of the microcracks is a result of the rapid solidification of partly or fully melted particles on the substrate [19]. The existence of various-shaped microsized particles on the surface when the particles hit the substrate could be related to the splashing of partly molten feedstock particles. The porosity of deposited coatings is related to the existence of coarse and fine pores. It was demonstrated that the coarse porosity in the plasma-sprayed coatings is attributed to structural defects. These defects were created by the incomplete filling of the voids between impacting powder particles to the surface [14,19]. SEM images of surface morphology demonstrated that the

addition of $TiO_2$ and $SiO_2$ into $Cr_2O_3$ powders resulted in the formation of a slightly lower amount of larger-sized voids on the surface.

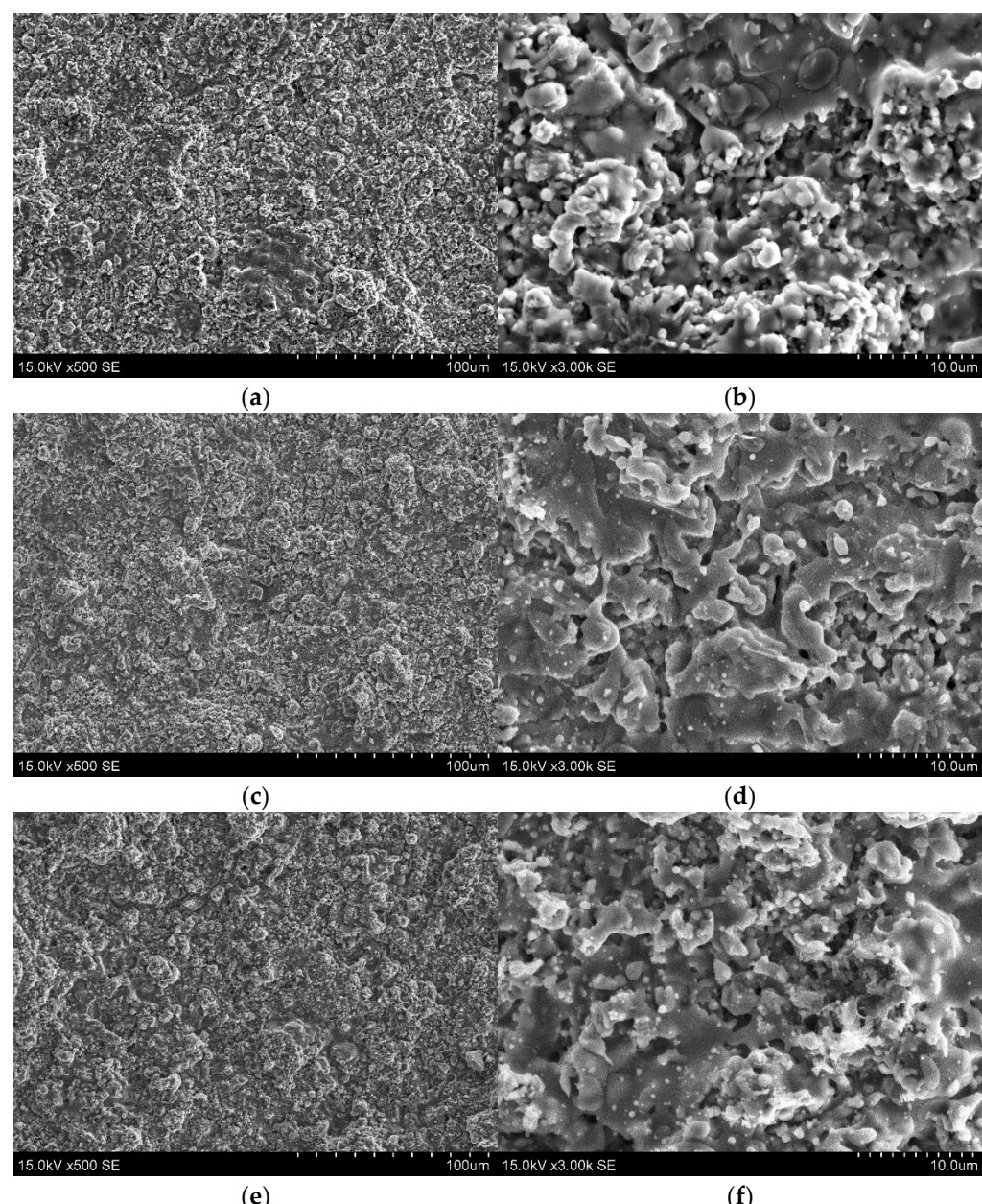

**Figure 1.** Surface morphology of (**a,b**) $Cr_2O_3$, (**c,d**) $Cr_2O_3$-$SiO_2$-$TiO_2$ and (**e,f**) $Cr_2O_3$-$SiO_2$-$TiO_2$-graphite coatings.

The distribution of chromium, oxygen, titanium, silicon and carbon on the $Cr_2O_3$-$SiO_2$-$TiO_2$ and $Cr_2O_3$-$SiO_2$-$TiO_2$-graphite coatings is presented in Figures 2 and 3. The EDS results showed that the distribution of chromium, oxygen and titanium on the surface of the $Cr_2O_3$-$SiO_2$-$TiO_2$ and $Cr_2O_3$-$SiO_2$-$TiO_2$-graphite coatings is homogenous. The EDS map image (Figure 3f) indicated that the graphite particles were quite homogeneously spread on the surface of the as-sprayed coating. In addition, single graphite particles with a size of up to 40 μm could be observed on the surface. It should be noted that silicon tends to agglomerate and form larger clusters on the surface of the chromia composite coatings (Figures 2f and 3e). The existence of larger-sized $SiO_2$ inclusions is due to the fact that the $SiO_2$ particles were not homogenously distributed in the raw powders and/or were agglomerated into bigger-sized clusters during manufacturing. The $Cr_2O_3$-$SiO_2$-$TiO_2$

coating was composed of chromium ($\sim$73.6 wt.%), oxygen (22.2 wt.%), titanium (3.3 wt.%) and silicon ($\sim$0.8 wt.%), and a low amount of carbon was obtained due to the absorption of atmospheric air. The as-sprayed coating consisted of Cr ($\sim$73.7 wt.%), O ($\sim$21.4 wt.%), Ti (3.0 wt.%), Si (0.8 wt.%) and carbon ($\sim$1.1 wt.%) when 10 wt.% of graphite was added to the initial $Cr_2O_3$-$SiO_2$-$TiO_2$ powders. The EDS measurements showed that the graphite content on the surface was up to 10 times lower compared to the feedstock powders. The chromia coating is composed of chromium (78.3 wt.%) and oxygen (21.3 wt.%), and a low amount of impurities (carbon and aluminum) was detected.

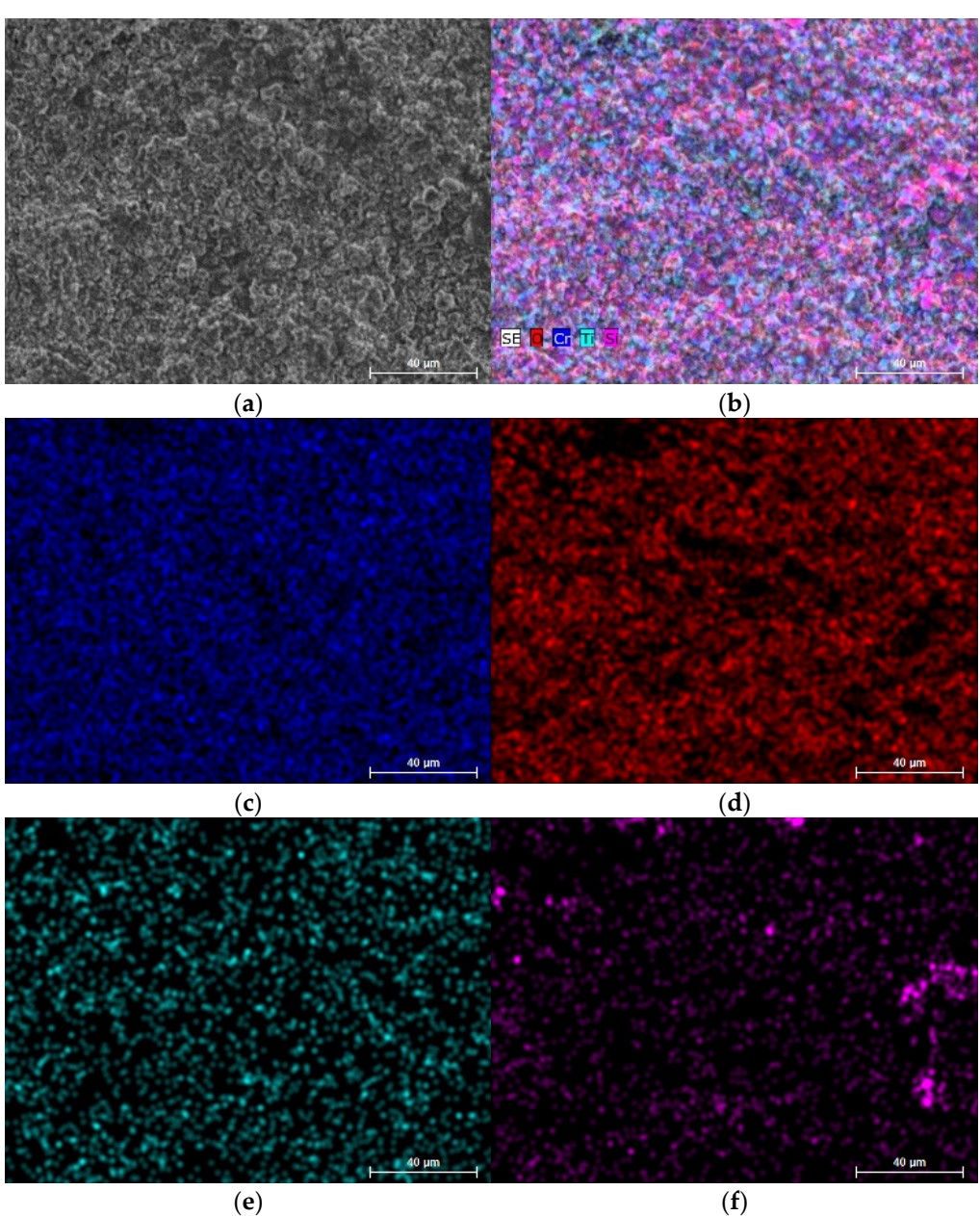

**Figure 2.** SEM micrograph (**a**) and elemental maps of $Cr_2O_3$-$SiO_2$-$TiO_2$ coating: (**b**) all elements. (**c**) chromium, (**d**) oxygen, (**e**) titanium and (**f**) silicon at 500 magnifications.

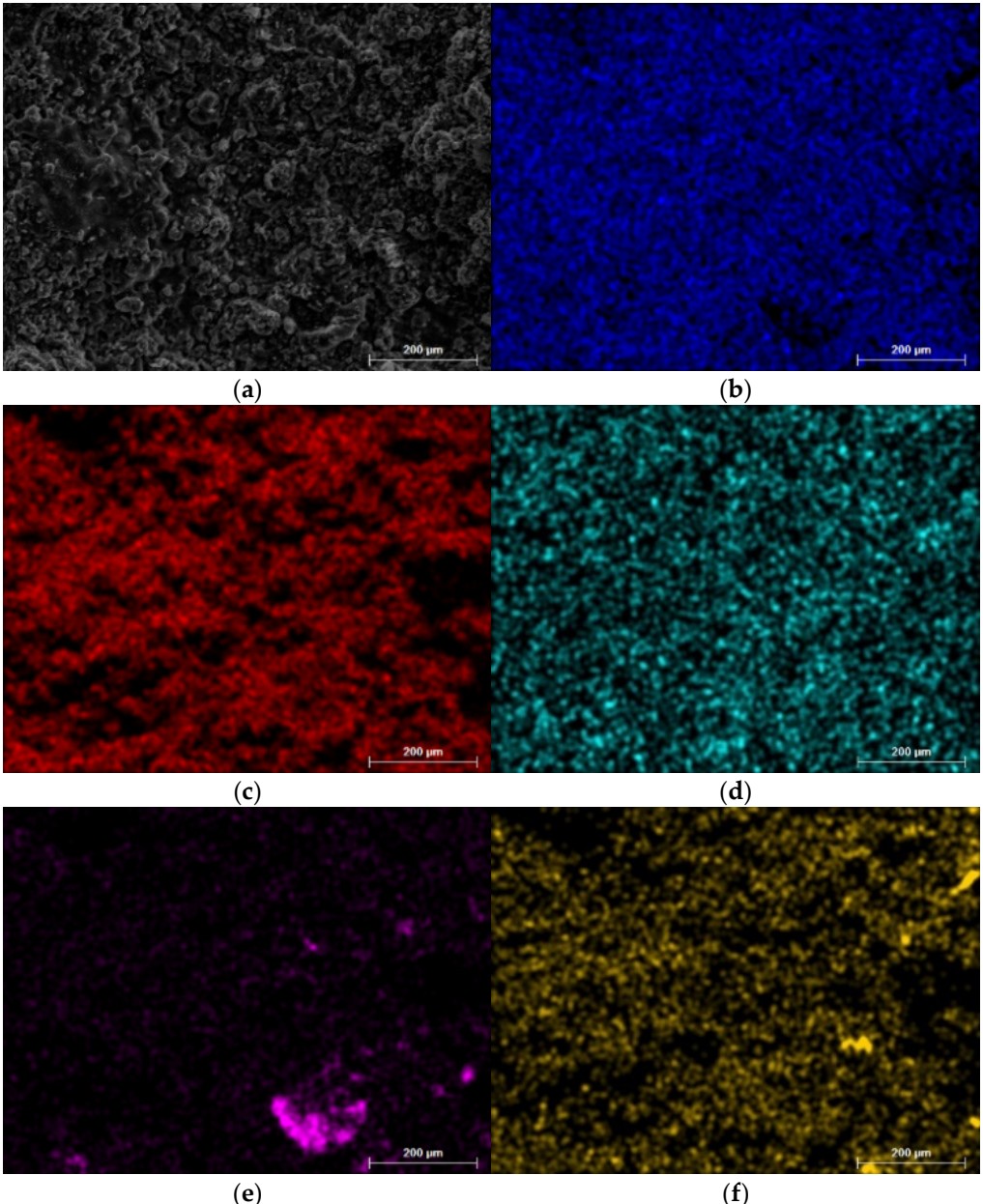

**Figure 3.** SEM surface image (**a**) and elemental maps of $Cr_2O_3$-$SiO_2$-$TiO_2$-graphite coating: (**b**) chromium, (**c**) oxygen, (**d**) titanium, (**e**) silicon and (**f**) carbon at 100 magnifications.

The surface roughness of the as-sprayed coatings is presented in Figure 4. The average surface roughness ($R_a$) and the root-mean-square roughness ($R_q$) of the $Cr_2O_3$ coating were ∼3.02 ± 0.51 µm and ∼3.77 ± 0.61 µm, respectively. The surface roughness of the $Cr_2O_3$-$SiO_2$-$TiO_2$ coating was slightly, up to ∼20%, higher. The $R_a$ value was 3.85 µm, while the $R_q$ value was 4.86 µm. Meanwhile, the roughness of the coating was drastically reduced (by 40%) when 10 wt.% of graphite was added to the $Cr_2O_3$-$SiO_2$-$TiO_2$ powders. The surface roughness $R_a$ and $R_q$ values for the $Cr_2O_3$-$SiO_2$-$TiO_2$-graphite coating were 2.22 µm and 2.77 µm, respectively. The reduction of surface roughness of the $Cr_2O_3$-$SiO_2$-$TiO_2$-graphite coating in comparison to the $Cr_2O_3$-$SiO_2$-$TiO_2$ coating could be related to the lower size and density of the graphite powders. If we compare the volume of both powders, there were almost equal parts of them in the feedstock mixture.

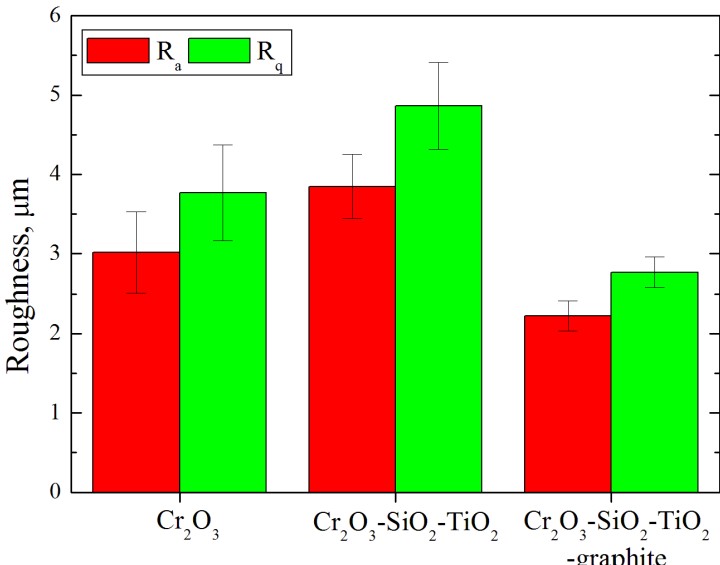

**Figure 4.** Surface roughness of as-sprayed coatings.

It was demonstrated that the addition of graphite nanoparticles (GNPs) or GNPs and carbon nanotubes (CNTs) into alumina reduced the surface roughness (up to 13%) of the as-sprayed alumina composite coatings [29]. Wang et al. [30] obtained that the addition of 3% or 6% of graphene into $Al_2O_3$-$TiO_2$ coatings reduced the porosity of the coatings. In addition, the surface of the coatings becomes flatter and denser after the addition of graphene. Similar results were found for the $Cr_2O_3$-$SiO_2$-$TiO_2$ coating with the addition of graphite into feedstock powders. The surface images of the $Cr_2O_3$-$SiO_2$-$TiO_2$ coating demonstrated that the amount of fully molten regions was higher compared to the $Cr_2O_3$ coating. However, the surface roughness of the $Cr_2O_3$-$SiO_2$-$TiO_2$ coating was enhanced. Zamani et al. [2] also obtained that the surface roughness of the $Cr_2O_3$ coating was lower compared to $Cr_2O_3$-$SiO_2$-$TiO_2$ coatings. However, the surface roughness was reduced with the increased amount of $Al_2O_3$ due to the lower melting temperature. The fraction of splats increases on the surface of deposited chromia or chromia composite coatings with the increase of the plasma jet temperature or supplementation of the metal oxides with lower melting temperature into feedstock powders [11,15,16,31].

The XRD graphs of the $Cr_2O_3$ and $Cr_2O_3$ composite coatings are given in Figure 5. It should be noted that the patterns of all coatings showed the same peaks, and only the intensities of the peaks are slightly different. The peaks are located at 24.6, 33.7, 36.3, 41.6, 50.3, 55.0, 63.6, 65.3 and 73.1 2θ degrees and are attributed to the (012), (104), (110), (202), (024), (116), (214), (300) and (1010) orientations of the $Cr_2O_3$ eskolaite phase, respectively [10,14,15,32]. The low- and broad-intensity peaks located at ~40.0°, ~44.4°, ~58.5°, and ~76.9° are assigned to the (006), (202), (122) and (220) planes of the eskolaite $Cr_2O_3$ phase, respectively [14,15,23]. The normalized intensities of the initial powders and as-sprayed coatings are presented in Table 1. It should be noted that the highest-intensity peak for the initial $Cr_2O_3$ powders was located at ~55.0°. Meanwhile, the as-sprayed chromia coatings had the highest peak at ~33.7°. The intensity of the peak at ~36.3° was slightly lower. It should be noted that the intensities of dominant peaks of the eskolaite $Cr_2O_3$ phase increased in the as-sprayed $Cr_2O_3$ coating (Table 1). The highest-intensity peak for the $Cr_2O_3$ coating was related to the (104) orientation (Figure 5). The addition of $SiO_2$-$TiO_2$ into the $Cr_2O_3$ coatings resulted in the intensity of the peak presented at ~33.7° (104) being slightly lower compared to the peak located at ~36.3° (110). The intensities of peaks for the $Cr_2O_3$-$SiO_2$-$TiO_2$-graphite and $Cr_2O_3$-$SiO_2$-$TiO_2$ coatings were similar (Figure 5). The highest-intensity peak in the $Cr_2O_3$-$SiO_2$-$TiO_2$ feedstock powders was obtained at ~36.3°. The order of the five highest peaks according to normalized intensity remains unchanged in the as-sprayed $Cr_2O_3$-$SiO_2$-$TiO_2$ coating. Only a slight increase or decrease in

the peak intensities was obtained (Table 1.). The addition of graphite into $Cr_2O_3$-$SiO_2$-$TiO_2$ resulted only in a slight reduction in the intensities of most peaks. However, the intensities of the peaks at ~41.6° and ~63.6° were enhanced for the $Cr_2O_3$-$SiO_2$-$TiO_2$-graphite coating.

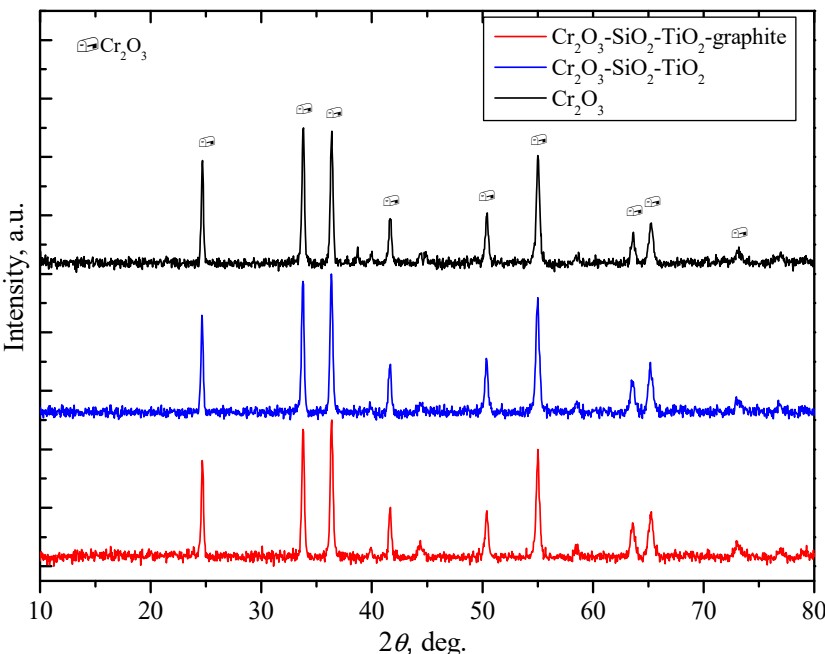

**Figure 5.** The XRD patterns of the $Cr_2O_3$, $Cr_2O_3$, $Cr_2O_3$-$SiO_2$-$TiO_2$ and $Cr_2O_3$-$SiO_2$-$TiO_2$-graphite coatings.

**Table 1.** The normalized intensity values of XRD peaks in the powders and coatings.

| Peak Position, º | Normalized Intensities Values | | | | |
| --- | --- | --- | --- | --- | --- |
| | $Cr_2O_3$ Powders | $Cr_2O_3$-$SiO_2$-$TiO_2$ Powders | $Cr_2O_3$ Coating | $Cr_2O_3$-$SiO_2$-$TiO_2$ Coating | $Cr_2O_3$-$SiO_2$-$TiO_2$-Graphite Coating |
| 24.6 | 0.446 [4] | 0.784 [4] | 0.763 [4] | 0.721 [4] | 0.710 [4] |
| 33.7 | 0.756 [2] | 0.933 [2] | 1.000 [1] | 0.947 [2] | 0.933 [2] |
| 36.3 | 0.500 [3] | 1.000 [1] | 0.955 [2] | 1.000 [1] | 1.000 [1] |
| 40.0 | 0.247 | 0.165 | 0.090 | 0.085 | 0.080 |
| 41.6 | 0.162 | 0.343 | 0.340 | 0.332 | 0.364 [5] |
| 50.3 | 0.392 [5] | 0.520 [5] | 0.360 [5] | 0.370 [5] | 0.337 |
| 55.0 | 1.000 [1] | 0.836 [3] | 0.795 [3] | 0.815 [3] | 0.780 [3] |
| 63.6 | 0.257 | 0.161 | 0.223 | 0.228 | 0.261 |
| 65.3 | 0.311 | 0.206 | 0.305 | 0.352 | 0.340 |

Superscripts indicate order of normalized peak intensities, lower numbers attributed to higher intensity.

The $TiO_2$ or $SiO_2$ phases were not detected in the XRD data of the $Cr_2O_3$-$SiO_2$-$TiO_2$ and $Cr_2O_3$-$SiO_2$-$TiO_2$-graphite coatings. Usually, the presence of the $TiO_2$ rutile phase in the XRD patterns of thermally sprayed coatings is attributed to peaks located at 27.5, 41.3, 54.4 and 56.7° [10,14,19]. However, it was observed that the $TiO_2$ rutile phase may not be detected in the XRD data of the chromia composite coatings with a low amount of $TiO_2$ (up to 5%) [32]. The crystalline graphite peak (at 26.5°) was not obtained in the XRD pattern of the $Cr_2O_3$-$SiO_2$-$TiO_2$-graphite coating [26,33]. Our previous study [26] demonstrated that the low-intensity crystalline graphite peak in alumina–graphite coatings was observed when the graphite content was 1.6 wt.%.

The variation of the friction coefficient curves depending on sliding time is given in Figure 6. Similar behavior of the friction coefficient curves was obtained for all as-sprayed coatings. The average values of the friction coefficient for the $Cr_2O_3$, $Cr_2O_3$-$SiO_2$-$TiO_2$ and $Cr_2O_3$-$SiO_2$-$TiO_2$-graphite coatings using 1 N load were estimated as ~0.369, ~0.316 and

~0.405, respectively (Figure 7a). The friction coefficient of the steel substrate was ~0.687 when 1 N load was used [15]. The friction coefficients of the coatings slightly increased with the enhancement of the applied load (Figure 7a). The lowest friction coefficient was obtained for the $Cr_2O_3$-$SiO_2$-$TiO_2$ coating and was ~0.378. Meanwhile, the highest value of the friction coefficient (~0.416) was observed for the $Cr_2O_3$-$SiO_2$-$TiO_2$-graphite coating. Thus, the friction coefficient slightly decreased (by 2%) when the $SiO_2$-$TiO_2$ additive was used. Meanwhile, the addition of $SiO_2$-$TiO_2$-graphite fractionally increased (up to 8%) the friction coefficient of the $Cr_2O_3$ composite coatings. Despite the fact that the friction coefficient of all the coatings increased with the increase in the applied load, it was significantly lower (up to 40%) than that of the P265GH steel (~0.630).

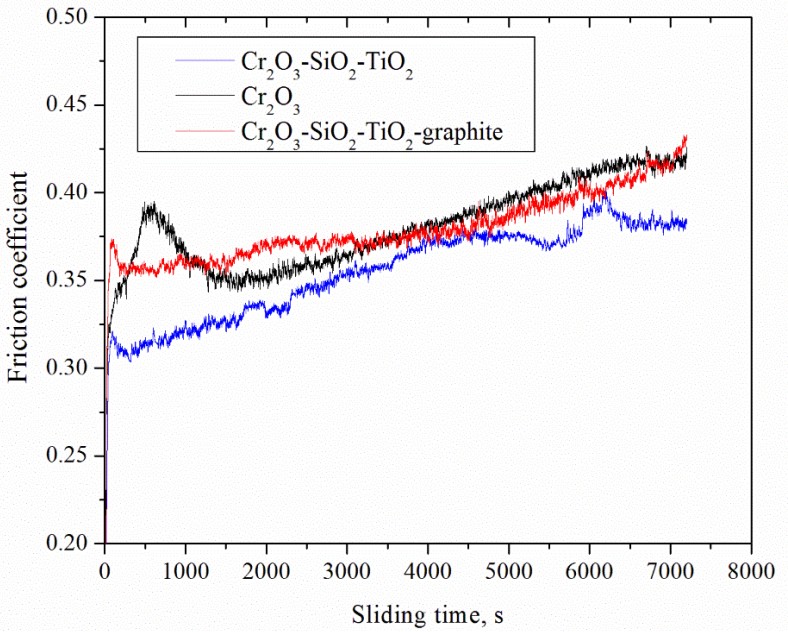

**Figure 6.** The variation of friction coefficients of coatings versus sliding time when 3 N load was used.

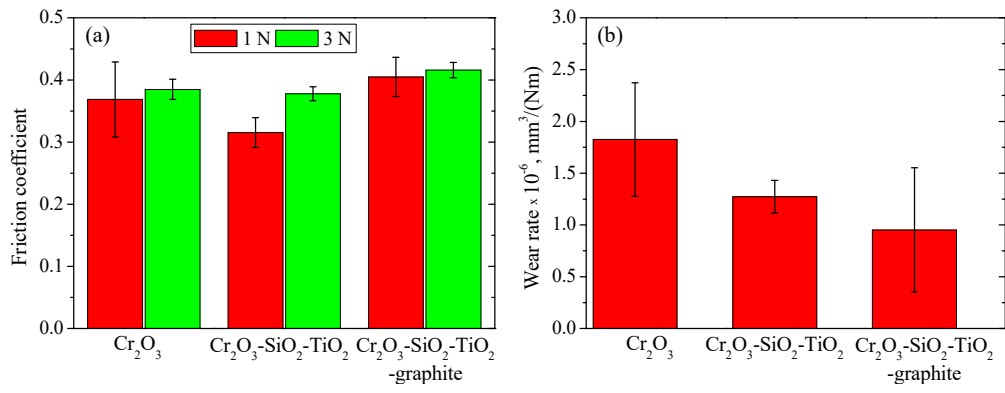

**Figure 7.** Friction coefficients of deposited coatings (**a**) and the specific wear rates of coatings when 3 N load was applied (**b**).

It was demonstrated that the addition of a low amount of graphene oxide could reduce the friction coefficient of alumina coatings by 15%. However, the friction coefficient could also be enhanced from 0.49 to 0.52 with the increase in the used load [34]. Priyadershini et al. [29] observed that the addition of a low amount of GNPs and CNTs into alumina coatings resulted in a higher friction coefficient at the initial (up to 300 m) sliding distances. It was obtained that the friction coefficient depends on the applied load, type of used counterpart, porosity, hardness, surface roughness of the material, etc. [11,14,21,34,35]. Liang et al. [35] observed that the average friction coefficient decreased with the increase in

the initial roughness values of the material. Riedl et al. [36] determined that the friction coefficient of $Al_2O_3$ coating with the alpha phase was slightly reduced after surface polishing. Meanwhile, the friction coefficient of the $Al_2O_3$ coating with the kappa phase demonstrated an opposite trend. The steady state of the friction coefficient for the $Cr_2O_3$ coating was reached after 1500 s of sliding time, while the steady state for the $Cr_2O_3$-$SiO_2$-$TiO_2$-graphite coating was reached after 200–300 s. The surface with a larger initial roughness will need more time to reach the steady friction coefficient values, because the sharp hilltops must be grounded off first [35,36]. In addition, the smoother the surface, the greater the contact area between the friction pair, which can lead to higher friction coefficient values. The slight increase in the friction coefficient versus time for ceramic coatings and ceramic-carbon-based coatings during the steady-state region was observed in several studies when dry-sliding and different loads were used [11,14,30,34].

The average specific wear rates of the deposited coatings using 3 N load are presented in Figure 7b. The wear rate of the chromium oxide coating was $\sim 1.83 \times 10^{-6}$ mm$^3$/(Nm). The specific wear rate decreased to $\sim 1.27 \times 10^{-6}$ mm$^3$/(Nm) with the addition of $SiO_2$-$TiO_2$ into the chromium oxide powders. The addition of the graphite in the feedstock powders stipulated the formation of the coating with the lowest specific wear rate of $\sim 0.95 \times 10^{-6}$ mm$^3$/(Nm). It should be mentioned that the specific wear rates of the P265GH steel were $6.68 \times 10^{-5}$ mm$^3$/(Nm) and $4.22 \times 10^{-5}$ mm$^3$/(Nm) when 1 N and 3 N loads were used, respectively. The reduction of the wear rate of the chromia composite coatings could be attributed to the enhanced densification, reduction of porosity, presence of transferred layer from the $Al_2O_3$ ball and formation of self-lubricating graphite layer (in the case of $Cr_2O_3$-$SiO_2$-$TiO_2$-graphite) [29,33]. The increase in the sliding wear rates of the $Cr_2O_3$ coatings with the incorporation of $TiO_2$ compared to the $Cr_2O_3$ coatings during the sliding tests against an $Al_2O_3$ counterpart was noticed by several authors [13,17,19,37,38]. It was obtained that the reinforcement of $TiO_2$ decreased the hardness, and as a result, the resistance to wear of the $Cr_2O_3$-$TiO_2$ coatings was reduced [13].

The wear tracks of the chromium oxide and its composite coatings under the loads of 1 N and 3 N are presented in Figure 8. It should be noted that the surface of the as-sprayed coatings was only slightly damaged when 1 N load was applied (Figure 8a,c,e). The randomly distributed worn areas of 10 μm to 30 μm in size could be seen on the surfaces. The surface profile measurements demonstrated similar profiles as for deposited coatings. Thus, it was not possible to calculate the wear rates of the as-sprayed coatings, as only the top hills were fractionally abraded when tribological tests were performed using a load of 1 N. Meanwhile, the specific wear rate of the steel substrate under 1 N load was $6.68 \times 10^{-5}$ mm$^3$/(Nm). Compared with 1 N, and with the increased load up to 3 N, the interfacial contact area of the coatings increased. The SEM images indicated that the wear scars became wider, brighter and more worn-out areas were observed (Figure 8). These results indicate that the damage level and wear loss increased and resulted in a higher wear rate of the coatings.

The distribution of chemical elements on the wear scars of the coatings after tribological tests using 3 N load is presented in Figures 9–11. A low amount of aluminum from the alumina ball was obtained on the worn scars of all coatings (Figures 9e, 10e and 11e). Additionally, due to the heat generated during the sliding process, the small concentrations of carbon at a contact zone were determined (Figures 9f and 10f). The EDS results showed that the concentration of chromium was ~71.6 wt.%, oxygen ~27.1 wt.%, aluminum 0.9 wt.% and carbon 0.4 wt.% on the wear tracks of the chromia coating after the tribological tests with 3 N load. The amount of Cr was ~67.7 wt.%, O—26.4 wt.%, Ti—3.0 wt.%, Si—1.5 wt.%, C—0.5 wt.% and Al—0.9 wt.% on surface of the $Cr_2O_3$–$SiO_2$-$TiO_2$ coating. The concentration of chromium was ~68.7 wt.%, oxygen ~25.4 wt.%, titanium ~2.7 wt.%, silicon ~0.7 wt.%, aluminum 1.1 wt.% and carbon 1.4 wt.% on the wear track of the $Cr_2O_3$–$SiO_2$-$TiO_2$-graphite coating. The increase in the oxygen concentration and the reduction of chromium amount on the wear scars for all coatings were observed after the tribological

tests. The slight oxidation of chromia ceramic coatings after tribological tests was observed by Amanov et al. [18] and is a result of abrasive wear.

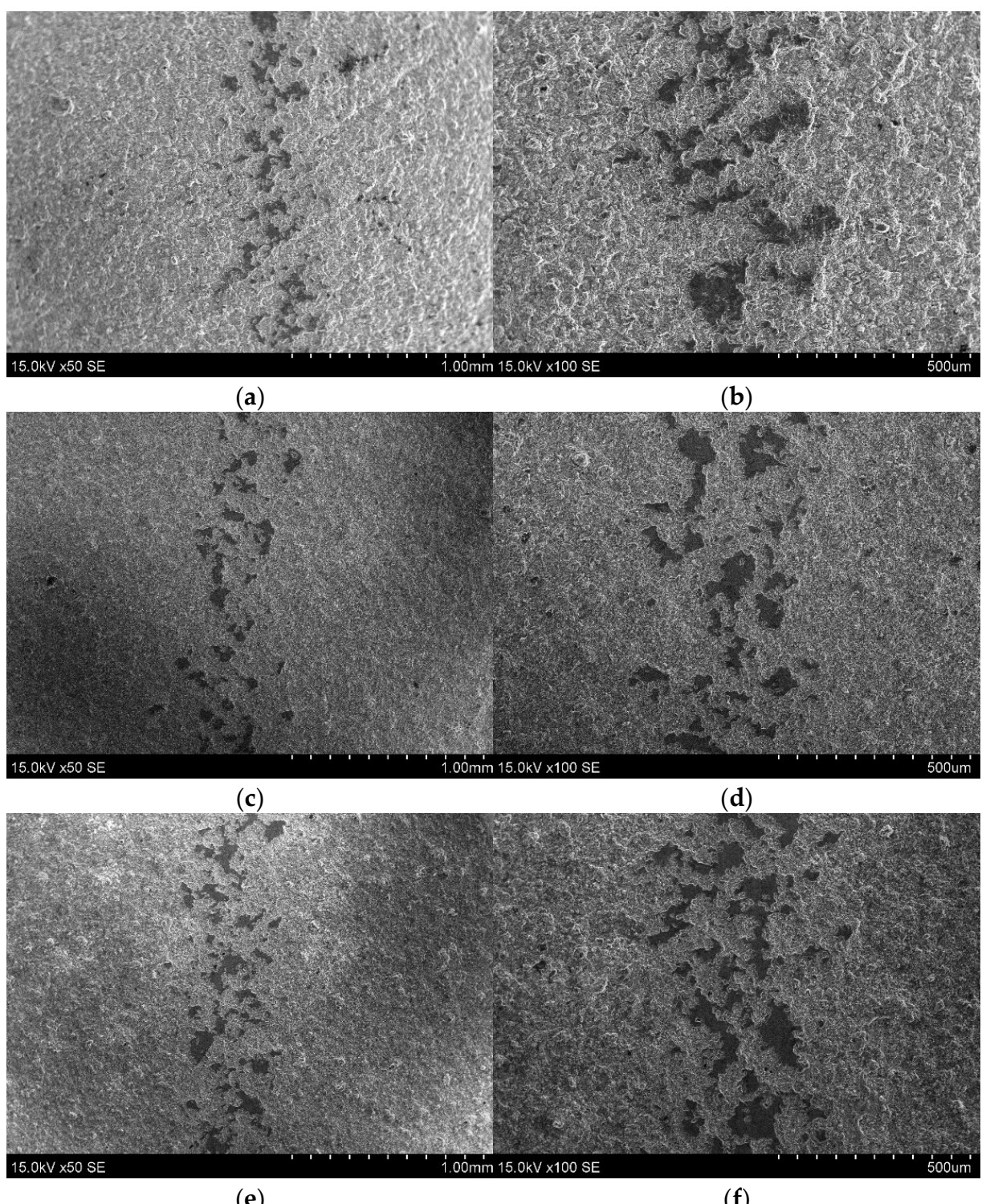

**Figure 8.** Wear scars views of (**a**,**b**) $Cr_2O_3$, (**c**,**d**) $Cr_2O_3$-$SiO_2$-$TiO_2$ and (**e**,**f**) $Cr_2O_3$-$SiO_2$-$TiO_2$-graphite coatings after tribological tests with (**a**,**c**,**e**) 1 N and (**b**,**d**,**f**) 3 N loads.

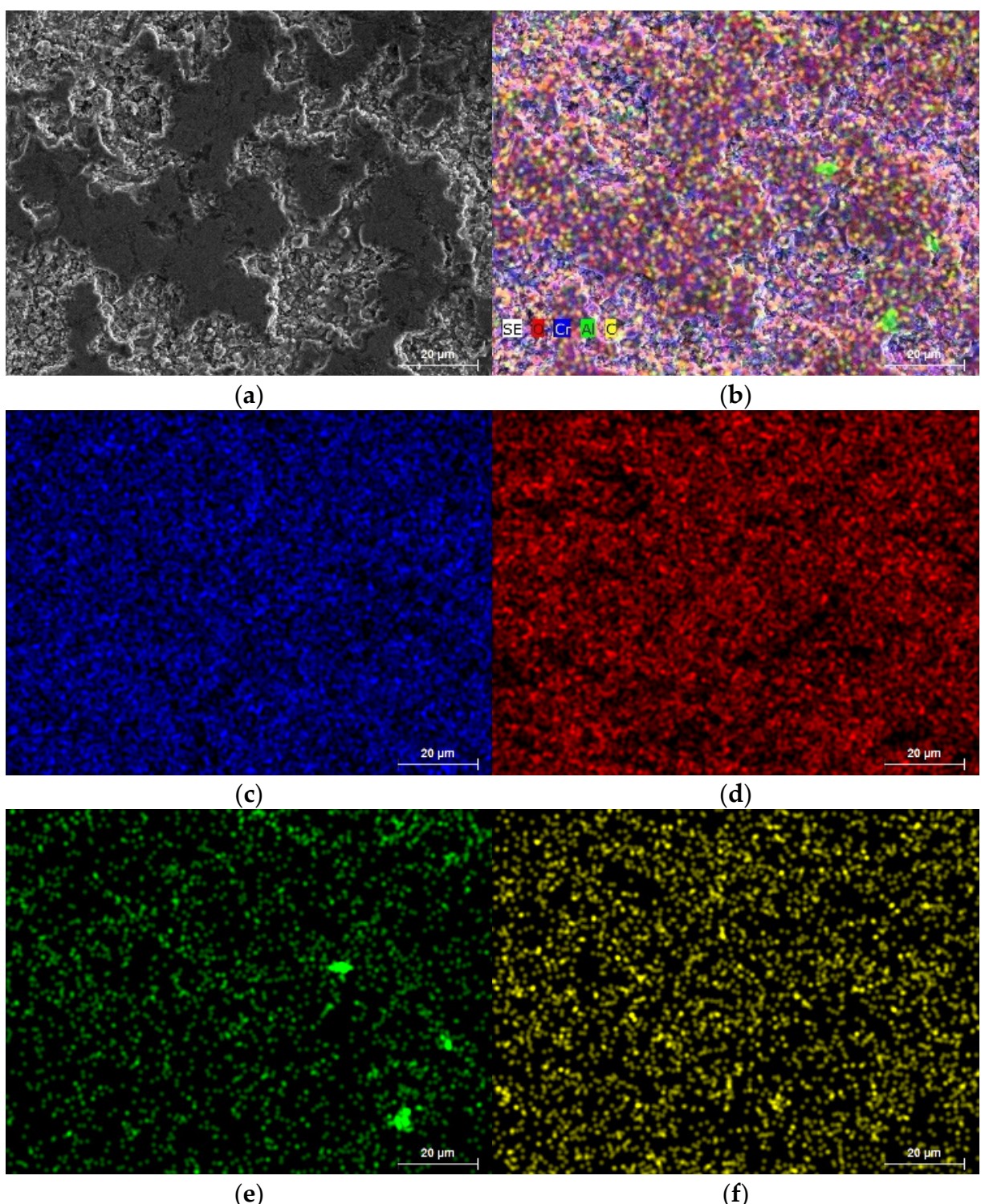

**Figure 9.** SEM (**a**) and EDS mapping images of worn surface of $Cr_2O_3$ coating, all detected elements (**b**),chromium (**c**), oxygen (**d**), aluminum (**e**) and carbon (**f**) on the wear scar surface of $Cr_2O_3$ coating.

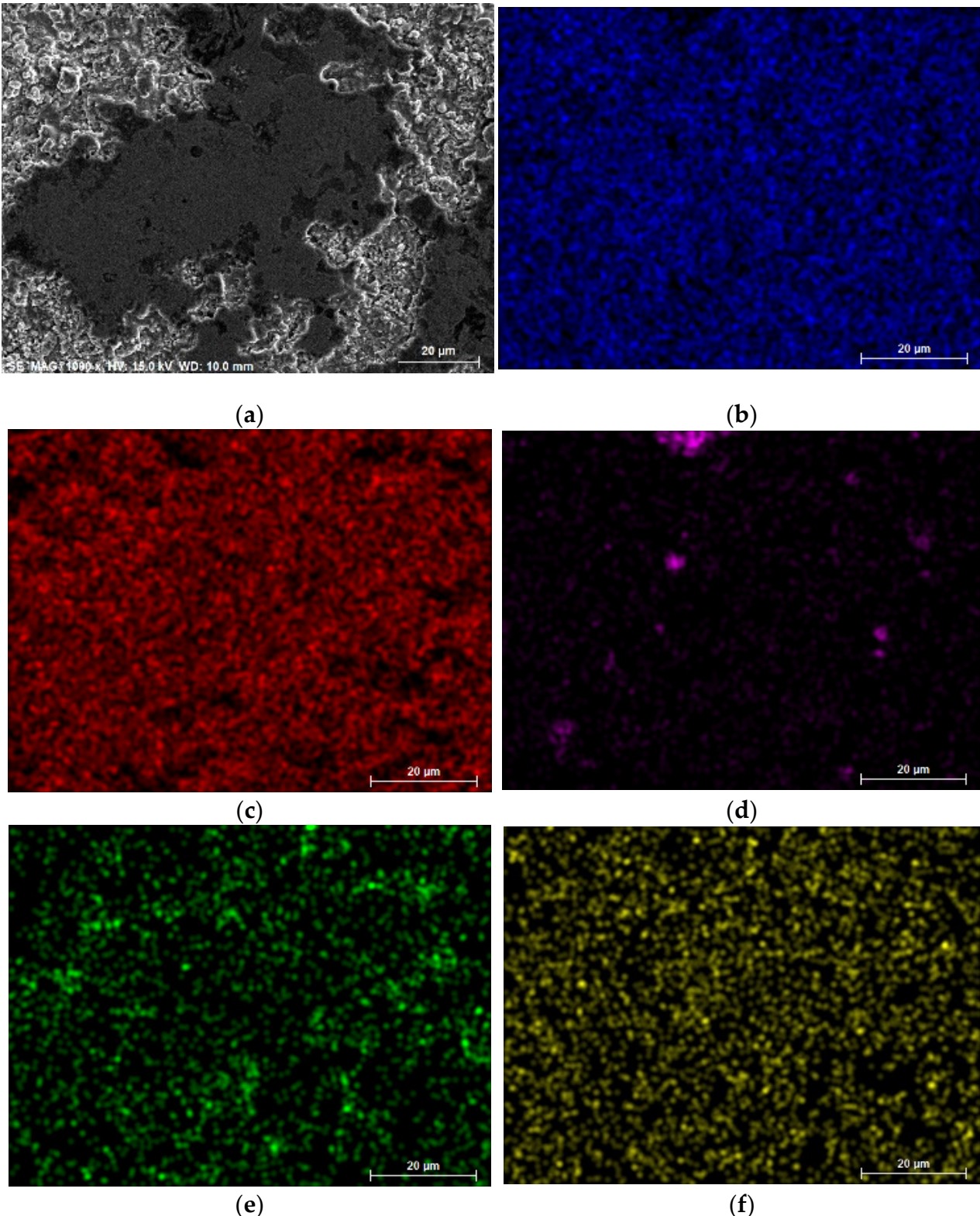

**Figure 10.** SEM (**a**) and EDS mapping images of worn surface of $Cr_2O_3$–$SiO_2$-$TiO_2$ coating, Cr (**b**), O (**c**), Si (**d**), Al (**e**) and C (**f**).

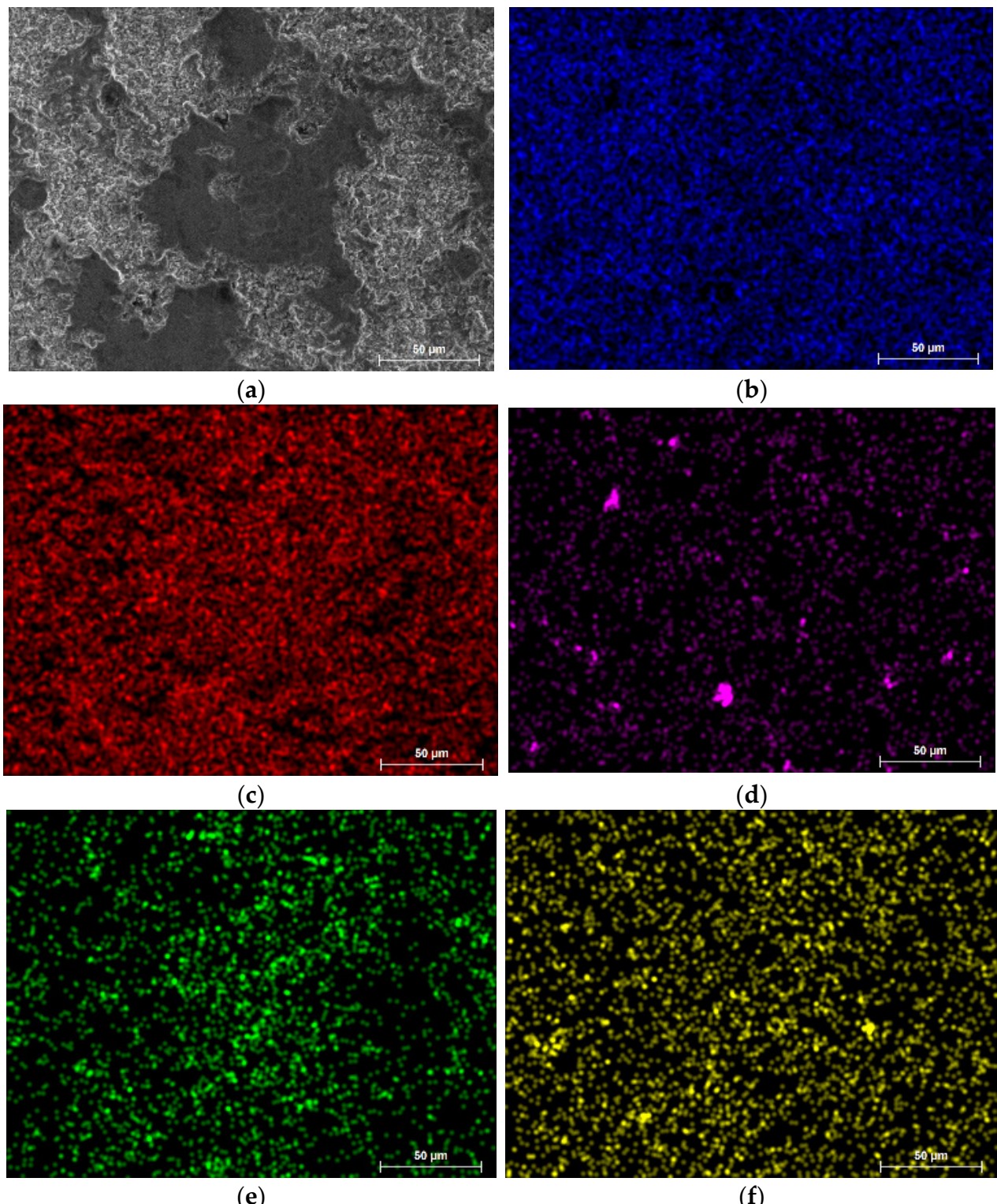

**Figure 11.** SEM (**a**) and EDS mapping images of worn surface of $Cr_2O_3$–$SiO_2$-$TiO_2$-graphite coating, Cr (**b**), O (**c**), Si (**d**), Al (**e**) and C (**f**) when 3 N load was used.

The EDS results indicated the presence of Al and an increase in carbon elemental concentration in the worn surfaces of the coatings after the tribological tests. A low amount of aluminum was (from 0.9 wt.% to 1.1 wt.%) observed for all coatings after the tribological tests. This confirms the formation of the tribolayer of Al and is a result of the wear between the counterpart ($Al_2O_3$ ball) and coating [34]. The existence of carbon

indicates the formation of a graphite-base film on the surface of the removed areas [26,33]. The graphite may act as a solid lubricant to prevent the direct contact between the $Al_2O_3$ ball and the surface of the coating. The SEM images indicated that the wear scars are noncontinuous (Figure 8). Thus, effective lubrication would be provided only in the direct contact zone. The EDS results and mapping images indicated that the amount of carbon at the wear scar areas increased from ~1.1 to ~1.4 wt.%. Feng et al. [33] demonstrated that the GNPs and graphite particles start to act as a self-lubricant, and a stable tribofilm layer is formed on the surface of the $Al_2O_3$-$TiO_2$ coatings when the applied loads are 15 N or higher. Thus, the friction coefficient values would remain similar or even higher compared to the $Cr_2O_3$-$SiO_2$-$TiO_2$ coating. In our case, the used 1 N and 3 N loads in tribological tests are considered as light loads. Thus, the wear volume was very low (see Figure 8), and the graphite particles could not be effectively pulled out from the $Cr_2O_3$-$SiO_2$-$TiO_2$-graphite coating to act as a self-lubricant.

Wear debris in the form of small microsized particles was obtained on the surfaces of the coatings. The size of the debris was very similar for all coatings. Usually, the interfacial bonding between the formed splat starts to delaminate at higher loads or longer sliding distances. However, the lamellar debris was not produced under used loads. Severe abrasive wear is probably the main wear mechanism during the dry-sliding tests. As for the single one, parallel grooves can be obtained on the wear tracks of the coatings [18,27,39,40]. The particles are extracted from the coating and are crushed in the contact zone between the surface of the coating and the sliding $Al_2O_3$ ball. The pulled-out $TiO_2$ or $TiO_2$-graphite particles start to form a lubricative layer and improve the tribological properties (especially wear rate). Meanwhile, the hard $Cr_2O_3$ particles act as an abrasive and can tear up the formation of the tribolayer. The traces of Al on the worn surfaces also indicate that the $Al_2O_3$ ball slightly rubs and wears and could be involved in the formation of the tribolayer. The increase in the graphite amount on the wear scars is evidence of the initial tribofilm formation. It was reported that the graphite layer led to the reduction of the shear stress level and plastic deformation in the contact zone, which reduces the wear rate of ceramic coatings [27,39].

The tribological investigations demonstrated that the addition of $SiO_2$-$TiO_2$ and $SiO_2$-$TiO_2$-graphite additives had an insignificant effect on the friction coefficient of the chromia coating when 3 N load was used. However, the wear resistance of the chromia composite coatings under dry-sliding conditions was improved by up to ~45% compared to the chromia coating.

## 4. Conclusions

$Cr_2O_3$ and chromia composite coatings were formed by atmospheric plasma spraying. The addition of $SiO_2$-$TiO_2$ enhanced (up to 20%) while the $SiO_2$-$TiO_2$ -graphite additive decreased (25%) the surface roughness compared to the $Cr_2O_3$ coating. The addition of graphite increased the homogeneity of the surface and enhanced the amount of molted areas on the surface of the $Cr_2O_3$–$SiO_2$-$TiO_2$-graphite coating. The concentration of carbon in the as-sprayed coating was only ~1.1 wt.%, indicating that most of the graphite was lost in the plasma flow due to sublimation. The XRD results indicated that the $Cr_2O_3$, $Cr_2O_3$–$SiO_2$-$TiO_2$ and $Cr_2O_3$–$SiO_2$-$TiO_2$-graphite coatings consisted only of the eskolaite phase. The lowest friction coefficient values of 0.316 and 0.378 were observed for the $Cr_2O_3$–$SiO_2$-$TiO_2$ coating when the friction tests were carried out using 1 N and 3 N, respectively. The amount of carbon on the wear scars was ~1.4 wt.%, which is ~25% higher compared to the as-sprayed coating. This indicates the formation of the self-lubricating layer on the surface. However, the applied loads were too low for the formation of the continuous wear tracks, and the effect of graphite as a solid lubricant was limited. The friction coefficient of the coatings was lower by 40% in comparison to the steel under dry-sliding conditions. The wear resistance of the $Cr_2O_3$ coatings was enhanced by ~30% and ~45% with the addition of $SiO_2$-$TiO_2$ and $SiO_2$-$TiO_2$-graphite, respectively. The improvement in tribological properties of the $Cr_2O_3$ composite coatings could be related to a higher amount of molted particle

areas on the surface due to the lower melting temperatures of $SiO_2$ and $TiO_2$. The lowest specific wear rate of $0.95 \times 10^{-6}$ mm$^3$/(Nm) was reached for the $Cr_2O_3$–$SiO_2$-$TiO_2$-graphite coating, which was considerably lower (up to 45 times) compared to the steel value.

**Author Contributions:** Conceptualization, L.B. and L.M.; methodology, L.B., M.M., R.K. and M.K.; software, L.B.; validation, L.B. and M.M.; formal analysis, L.B. and L.M.; investigation, L.B., L.M., M.M., R.K. and M.K.; data curation, L.B.; writing—original draft preparation, L.B., M.M. and L.M.; writing—review and editing, L.B. and L.M.; visualization, L.B. and L.M.; supervision, L.B.; All authors have read and agreed to the published version of the manuscript.

**Funding:** This research received no external funding.

**Institutional Review Board Statement:** Not applicable.

**Informed Consent Statement:** Not applicable.

**Data Availability Statement:** Not applicable.

**Acknowledgments:** The authors thank M. Grigaliūnas from the Kaunas University of Technology and S. Matkovič from the University of Ljubljana for tribological measurements.

**Conflicts of Interest:** The authors declare no conflict of interest.

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
