# Peer review of "Tribological Properties of Cr2O3, Cr2O3–SiO2-TiO2 and Cr2O3–SiO2-TiO2-Graphite Coatings Deposited by Atmospheric Plasma Spraying"

_coatings, doi:10.3390/coatings13020408_

Round 1
Reviewer 1 Report
Review of the manuscript ‘Tribological properties of Cr2O3, Cr2O3–SiO2-TiO2 and Cr2O3–SiO2-TiO2-graphite coatings deposited by atmospheric plasma spraying’ by Lukas Bastakys, Liutauras Marcinauskas, Mindaugas Milieška, Mitjan Kalin and Romualdas Kėželis.
Comments to the Authors
The paper describes the tribological properties of Cr2O3 coatings doped with SiO2 and graphite. This paper requires corrections and after that is publishable in Coatings. However, in light of the following concerns, I recommend making major modifications to this paper before reconsidering.
The reviewer would like to address the following issues:
1. The author should clarify the importance and novelty of this work.
2. The authors should add a larger scale bar in every picture. In Figs. 2, 3, 9, 10 and 11 scale bars are too small.
3. In Fig. 5 (XRD results) are some peaks at 40 °, 45 °, 58 °, and 77 °. Authors should check it and describe it.
4. Why did the authors use those parameters in wear tests? Maybe the test should last longer because in Fig. 6 the friction coefficient keeps increasing.
5. The authors should investigate the wear properties of P265GH steel. This would give information about whetherthe coatings are beneficial or not.
6. What about the third body? In the opinion of the reviewer, during the wear test, the authors generate a lot of third body, and it is necessary to investigate that.
7. In Fig. 7a, the friction coefficient after wear test at 1N and 3N is presented, but in Fig. 7b, the specific wear rate for samples tested with the load equal to 3N are shown. Why?
8. The results of the SEM and EDS analysis are presented only for the sample tested with 3N load. Why?
9. The quality of Figs. 9-11 is poor. Please upload the pictures in higher resolution.
Besides the aforementioned, the reviewer advised to check the language, maybe by a certified professional.
Reviewer 2 Report
In this paper, the Cr2O3, Cr2O3-SiO2-TiO2 and Cr2O3-SiO2-TiO2-graphite coatings were formed by atmospheric plasma spraying. The influence of SiO2-TiO2 and SiO2-TiO2-graphite rein-forcements on the surface morphology, elemental composition, structure and tribological properties of chromia coatings were researched. There are some detailed suggestions as follows:
Major problems:
1 The main aim of this paper was to study the effect of the addition of SiO2-TiO2 and SiO2-TiO2-graphite on the surface morphology, phase structure, elemental composition and tribo-logical properties under dry sliding of chromia composite coatings deposited by atmos-pheric plasma spraying. However, the relevant pictures of the test device were not provided, and the size of the final sample is not clearly written (40X10X6 mm?). It is mentioned that two samples were used for tests of each kind of coating, but we usually use three or five samples to avoid the influence of human and uncertain factors on the test results. Two is not enough.
2 The writing of conclusions needs to be improved. It is suggested to write one by one according to the content, i.e.,
1)............;
2).............
Other problems:
1. What is the meaning of "GNP" and "CNT" in the first paragraph of page 6? I can guess their meaning, but please provide the full name when it first appears.
2. Please revise the format of references according to the latest published manuscript.
3. Please check the writing form of numbers in the chemical formulas in the reference.
4. The expression of friction coefficient should be uniform in the full text, coefficient of friction or friction coefficient. Coefficient of friction is better.
Based on the above problems, this paper isn't suitable to publish in its current vision. Minor revision is my opinion.
Reviewer 3 Report
The article Tribological properties of Cr2O3, Cr2O3–SiO2-TiO2 and Cr2O3–SiO2-TiO2-graphite coatings deposited by atmospheric plasma spraying presents the results of studying the properties of various coatings obtained by plasma spraying, as well as their characterization using various analysis methods. In general, this line of research is very relevant and appropriate for further practical application. The presented results correspond to the topics of the proposed journal, however, before this article is accepted for publication, the authors should answer all the questions of the reviewer.
1. The authors present data on the morphological features of the coatings obtained, while according to the data presented, the coatings obtained have a very developed surface morphology, in this regard, the authors should reflect how exactly the surface morphology was taken into account when considering the tribological characteristics.
2. According to the presented mapping data, the authors show the presence of silicon inclusions, which are located in the form of separate grains or inclusions, the authors should give a more detailed explanation of the presence of such inclusions and their distribution isotropy.
3. The roughness of these coatings is quite different, and the authors should determine exactly how it was determined.
4. X-ray diffraction data for different types of coatings as presented do not reflect the general changes depending on the different compositions. In this case, the authors should provide data on the ratio of phases and their change depending on the composition.
5. The change in the coefficient of dry friction, shown in Figure 6, requires additional explanation and an explanation of how exactly the morphological features of the surface of the coatings are taken into account?
6. How the changes in the coefficient of dry friction were determined with the presented error, which is quite significant.
Round 2
Reviewer 1 Report
Thank you for the answers.
Reviewer 3 Report
The authors answered all the questions, the article can be accepted for publication.